# Ketamine in Acute Brain Injury: Current Opinion Following Cerebral Circulation and Electrical Activity

**DOI:** 10.3390/healthcare10030566

**Published:** 2022-03-17

**Authors:** Christian Zanza, Fabio Piccolella, Fabrizio Racca, Tatsiana Romenskaya, Yaroslava Longhitano, Francesco Franceschi, Gabriele Savioli, Giuseppe Bertozzi, Stefania De Simone, Luigi Cipolloni, Raffaele La Russa

**Affiliations:** 1Department of Emergency Medicine, Policlinico Gemelli/IRCCS, University of Catholic of Sacred Heart, 00168 Rome, Italy; christian.zanza@live.it (C.Z.); francesco.franceschi@unicatt.it (F.F.); 2Department of Anesthesia and Critical Care Medicine, Azienda Ospedaliera “SS Antonio e Biagio e C. Arrigo”, 15121 Alessandria, Italy; fpiccolella@ospedale.al.it (F.P.); fracca@ospedale.al.it (F.R.); tatsiana_romenskaya@yahoo.it (T.R.); lon.yaro@gmail.com (Y.L.); 3Department of Emergency Medicine, Anesthesia and Critical Care Medicine, Michele and Pietro Ferrero Hospital, 12060 Verduno, Italy; 4Emergency Department, Fondazione IRCCS Policlinico San Matteo, 27100 Pavia, Italy; gabrielesavioli@gmail.com; 5Department of Clinical-Surgical, Diagnostic and Pediatric Sciences, School in Experimental Medicine, University of Pavia, 27100 Pavia, Italy; 6Section of Legal Medicine, Department of Clinical and Experimental Medicine, University of Foggia, 71122 Foggia, Italy; giuseppe.bertozzi@unifg.it (G.B.); stefdesim@gmail.com (S.D.S.); raffaele.larussa@unifg.it (R.L.R.)

**Keywords:** ketamine, acute brain injury, trauma brain injury, cerebral electrical activity, intracranial pressure, cerebral circulation

## Abstract

The use of ketamine in patients with TBI has often been argued due to its possible deleterious effects on cerebral circulation and perfusion. Early studies suggested that ketamine could increase intracranial pressure, decreasing cerebral perfusion pressure and thereby reducing oxygen supply to the damaged cerebral cortex. Some recent studies have refuted these conclusions relating to the role of ketamine, especially in patients with TBI, showing that ketamine should be the first-choice drug in this type of patient at induction. Our narrative review collects evidence on ketamine’s use in patients with TBI. Databases were examined for studies in which ketamine had been used in acute traumatic brain injury (TBI). The outcomes considered in this narrative review were: mortality of patients with TBI; impact on intracranial pressure and cerebral perfusion pressure; blood pressure and heart rate values; depolarization rate; and preserved neurological functions. 11 recent studies passed inclusion and exclusion criteria and were included in this review. Despite all the benefits reported in the literature, the use of ketamine in patients with brain injury still appears to be limited. A slight increase in intracranial pressure was found in only two studies, while two smaller studies showed a reduction in intracranial pressure after ketamine administration. There was no evidence of harm from the ketamine’s use in patients with TBI.

## 1. Introduction

Ketamine is a non-competitive antagonist of the NDMA receptor, discovered in 1956 with promising general anesthetic properties [1]. After nine years and still today, ketamine got in the operative room as a general anesthetic in humans. Chemically, ketamine is called 2-(2-chlorfelin)-2-(methylamino) cyclohexanone with two isomeric forms: (S)-(+) and (R)-(−). Currently, S-ketamine, the more potent of the two stereoisomers, is the most available or racemic mixture, containing both the (S+) and (R+) forms.

In addition to being NMDA antagonist, ketamine also shows synergism with other receptors such as opioid, monoaminergic, cholinergic, nicotinergic, and muscarinic, conferring a broad neuro-pharmacological pleiotropism [2].

After binding with the NMDA receptor, the main clinical effect is dissociative without vascular bed dilatation, so that no hypotension or heart rate variation is induced. This clinical combination makes ketamine potentially attractive when hypotension and cardiogenic shock occur, or high perfusion pressure is a priority, such as in patients with TBI [2,3].

This last clinical condition had been debated for many years; indeed, in the 1990s, ketamine was abandoned, on the assumption that the drug negatively influenced intracranial pressure [4].

Even more, its use must be carefully evaluated in cases of traumatic brain injury (TBI). In these cases, various modifications occur in the damaged brain tissue [5,6,7,8], down to the subcellular level [9,10,11,12]. However, the greatest caution concerning the use of ketamine in patients with TBI should be linked to the potential increase in intracranial pressure through sympathetic stimulation, worsening the outcomes. Nevertheless, it has been observed that if combined with γ-aminobutyric acid (GABA), ketamine does not raise intracranial pressure [13].

Moreover, spreading depolarization (SD), an anomalous propagation of electrical activity described by electroencephalogram (EEG) or by internal measurement by adequate electrodes, is associated with worsening outcomes in patients with TBI [14,15,16]. It is a near-complete disruption of the transmembrane ion gradient, which takes origin from areas of local acute ischemia, as an expression of tissue suffering from lack of energy [17].

Recently, some trials have shown that a curative effect of ketamine may occur after TBI, probably related to the suppression of SD onset after brain damage [18]. Furthermore, the outcomes analyzed in the present literature review were: mortality of patients with TBI; impact on intracranial pressure and cerebral perfusion pressure; blood pressure and heart rate values; depolarization rate; preserved neurological functions. This would allow, far from being considered a best clinical practice, both to understand the risks of a routine application of this substance in a clinical governance perspective and to lay the scientific bases to justify its use in a medico-legal context.

## 2. Materials and Methods

The systematic review was performed following the Preferred Reporting Items for Systematic Review and Meta-Analyzes (PRISMA) guidelines [19] analyzing titles, abstracts, and full text with Covidence online software [20]. Keywords were anesthetics, Ketamine, Traumatic brain injury, and Intracranial pressure.

The only papers excluded were narrative reviews, abstracts, letters, case reports. Another exclusion criterion was the administration in a non-comparable study population (elective surgery patients, abdominal surgery patients, or cardiac surgery patients).

Of 625 studies, only 53 met inclusion criteria. Among them, 11 recent studies passed inclusion and exclusion criteria and were included in this review.

Thus, the main selection bias is the wide temporal distribution of the articles (1981–2019), with an enormous variability between populations and types of invasive and non-invasive therapies.

## 3. Results and Discussion

Table 1 and Table 2 summarize the main results from the literature review and main ketamine effects from the systematic review.

### 3.1. Human Cerebral Circulation: Intracranial Pressure, Cerebral Perfusion Pressure, Mean Arterial Pressure, Heart Rate

Ketamine is a medicine with multiple effects that are applied in neurological/neurosurgical diseases. The principal mechanism of ketamine is NMDA receptor antagonism which then leads to the inhibition of glutamate activation. This inhibition leads to the suppression of the activity of the sensory cortex, limbic system, and thalamus, thus promoting the effect of dissociative anesthesia. At the peripheral level, ketamine acts on NMDA receptors, supporting the pain relief mechanism [21].

Seven studies analyzed the impact on intracranial pressure (ICP) in adults [23,24,25,28,29,30,31] and one in the pediatric population [25] with TBI, subarachnoid hemorrhage, or other intra-cranial traumatic diseases who had been admitted to intensive care in mechanical ventilation.

In three studies, no differences were detected regarding ICP between patients who received ketamine and the control group, and none showed pathological ICP [23,27,28,29,30]. Additionally, ketamine was found to prevent cough reflexes in one study [52]. After ketamine bolus, one study showed decreased ICP values in a short time and increased values over a long time without evidence of threat [24]. The pediatric population study, always after two minutes from ketamine bolus, also showed a decrease in median 7.8 mmHg in ICP [25].

A potential bias of this study was that 66% of the analyzed population received hyperosmolar fluids (mannitol or 3% NaCl before ketamine infusion). A notable common positive outcome was that of seven papers selected, none reported adverse events related to elevated ICP or rising mortality rates after ketamine’s use in this type of patients [23,24,27,28,29,30].

Cerebral perfusion pressure (CPP) monitoring was included in the previous studies [23,24,25,27,28,29,30,31]. Only one [25] reported an increased CPP of 3.9 mmHg for two minutes after ketamine bolus without adverse effects.

MAP (mean arterial pressure) and HR (heart rate) were evaluated in eight studies [22,23,24,26,27,28,29,30] and in six there were no differences, instead of in one paper [25] an increase in MAP after ketamine was reported before stressful maneuvers. Kolenda et al. showed an increase in HR of 20 beats/min on days 2, 3, 7 after ketamine administration [28].

Moreover, the systemic review published by Zeiler et al. [27] confirmed that ketamine leads to no elevation of intracranial pressure in austere TBI in either intubated or nonintubated patients. Ketamine use also leads to an increase in CPP and consequently to a decrease in the dosages of vasopressors necessary to counteract the effect of opioid-based sedatives. The grade of evidence for this affirmation is Oxford 2b, GRADE C recommendation. In addition, when used as a continuous infusion, the efficacy of ketamine is comparable to opioid-based sedation [29,32].

### 3.2. Ketamine, Spreading Depolarization and Burst Suppression

It has been observed that ketamine doses influence the electroencephalogram (EEG) tracing in a dose-dependent manner. Akeiju et al. [33] and Vlisides et al. [34], in their research, demonstrated that at the standard ketamine dosage required to induce unconsciousness, TBI EEGs showed a “gamma burst” pattern consisting of alterations in slow delta waves and gamma waves, associated with an increase in theta waves and a decrease in alpha and beta waves. In addition, in one study a quantitative EEG was used to determine deep sedation induced by ketamine, which may subsequently worsen the outcome of TBI [35].

Four studies [18,23,25,31] evaluated SD and burst suppression (EEG activity) during ketamine administration in brain-damaged patients. One of two studies, performed by the same team, concluded that ketamine significantly reduces SD, with a dose-dependent mechanism [17,18].

Instead, Hurtle et al., in their multicenter retrospective study [25], evaluated the electro-cortical graphic waves directly associated with brain activity (alpha, beta, delta, and theta (ECo G): in 43 patients, these waves and their mutual proportions were a potential predictor of SD. Primarily, this study showed the inversely proportional ratio between beta frequencies decreasing and a consensual increase of SDs number [25]. Secondly, a beta frequency increase corresponds to a reduction in SD and, since ketamine increases the presence of beta frequencies, it consequently limits the appearance of SD [25]. As far as the other waves (alpha, delta, and theta), no association was detected [25]. Moreover, Albanse et al. [24] showed the appearance of burst suppression with a dose-dependent relationship on the EEG of patients treated with ketamine. A bias of this last study was that EEG was performed with external electrodes.

A small pilot randomized study that recruited 10 patients with severe traumatic brain injury (TBI) or aneurysmal subarachnoid hemorrhage (SAH) was conducted by Carlson A. et al. [32]. The authors concluded that ketamine provides effective inhibition of SD, emphasizing that SD plays an important role in delayed ischemia and secondary brain damage. This randomized study further demonstrated a dose-dependent inhibitory effect of ketamine for the incidence of SD in case the patient received ketamine at a dosage below 1.15 mg/kg/h or was not being treated with ketamine at all [32].

Furthermore, it has been shown that ketamine can partially favor the increase in beta waves and consequently impede the appearance of SD. In any case, it was a retrospective study with only a limited effect of ketamine on beta frequencies and SD, indicating that additional prospective studies would be desirable. External EEG recordings also highlighted the appearance of burst suppression as a consequence of ketamine administration, accordant the theory of an inhibitory effect on the neuronal activity of ketamine [24].

### 3.3. Ketamine Dosage

Fatal effects or life-threatening side effects were not reported and, in all papers, analyzing the ketamine bolus was not used uniformly and the dosage was 1–5 mg/kg [24,25,31]. In the outstanding research instead, the dosage of continuous intravenous administration was 0.3–200 mg/kg/h [18,23,26,27,28,29,30,31,32,33,34,35,36].

Four studies [23,29,30,31] indicated precise ketamine dosage titlated to the desired sedation level according to Ramsey Score or the Riker sedation-agitation scale. It is strongly recommended to keep the RASS score between -3 and -4 initially, which is then modified according to the therapeutic objectives [16].

The dosage of ketamine varies widely between studies and in many, there were concomitant medications (propofol, fentanyl, sufentanyl, midazolam, morphine, and etomidate) which can mask the real effects. The most anesthetic drug used in the operating room and ICU is propofol which among the anesthetics is the one that reduces the ICP quickly more than the others, in patients with TBI [37]. Moreover, the ketamine appears to have different effects if carried out before, during, and after an experimentally induced head injury as analyzed [38].

### 3.4. Controlled Ventilation and Arterial CO_2_

Related to its pharmacokinetics, pharmacodynamics, and central nervous system effect, ketamine is a potential alternative to be considered in TBI patients who require mechanical ventilation or in combination with other sedatives. In addition, one of the effects of ketamine is vasodilation and bronchodilatation [13,39]. Thus, ketamine is strongly recommended in patients with severe TBI who have asthma and/or chronic obstructive pulmonary disease (COPD) or situations at risk for severe bronchospasm [35,39]. This makes ketamine a useful drug to be used in TBI situations where it is necessary to maintain hemodynamic stability and avoid respiratory depression.

Furthermore, some authors reported that ketamine should be considered one of the best agents to facilitate airway management in patients with traumatic brain injury [40]. Even though ventilation management has a pivotal role in patients with severe head injuries [41], none of the 11 studies reported the monitoring of arterial CO_2_. On the other hand, in all studies, patients received mechanical ventilation, with no patient in spontaneous breath, which, in animal models, was related to increases in ICP during ketamine sedation [42,43].

We must therefore recommend caution in reaching any conclusion regarding potential link between ketamine’s intake subsequent effects on ICP levels. Unregulated mechanical ventilation in the absence of PaCO_2_ control can create bias and confusion. A meta-analysis by Zeiler FA. et al. has shown that patients receiving a bolus of ketamine in addition to the baseline sedative plan and with pCO_2_ well-controlled by the ventilator, had no impact on ICP and therefore can be considered safe in this patient [28].

### 3.5. Ketamine and Children

Only one study was identified to evaluate ketamine administration in children [24]. In this study, the authors found a decrease in ICP and an increase in CPP immediately after administration [26,45].

However, as in many other studies, no data on mechanical ventilation as well as blood gas analytics were produced. In addition, 2/3 of patients were treated with a hyperosmolar solution before ketamine administration, making it difficult to estimate the ketamine outgrowth [25]. Efforts have been spent to understand if the administration of anesthetics, in general, can affect brain development in humans [46]. Recently, a systematic review [45] on the effects of general anesthetics on children was published.

Based on the literature review analysis, as it stands, no prospective or retrospective studies have been conducted involving children to study long-term outcomes using ketamine as an anesthetic agent.

Reached from current knowledge, therefore, the use of ketamine should be approached with caution in children due to the limited amount of data available.

### 3.6. Ketamine Toxicity

Regardless of the various possible and useful therapeutic applications of ketamine, its psychotropic properties on the CNS, well known in the forensic toxicological field, limit its widespread clinical use [53].

According to Krystal et al., indeed, even at 0.1 mg/kg doses, subjects may experience “endogenous psychosis-like” symptoms, behaviors, and cognitive deficits [47]. It should be remembered that ketamine use induces a state, known as “dissociative anesthesia”, in which the individual, albeit cardio-pulmonary functioning, is unable to respond to sensory stimuli [1]. These patterned effects, characterized by dissociation with visual, auditory, and somatosensory hallucinations and space–time distortion, have made ketamine a popular recreational drug. On the other hand, at higher doses, these effects will be amplified inducing a schizophrenia-like clinical condition [47,48]. Although these effects are transient and reversible, long-term use can cause cognitive impairment with severe cerebral atrophy [50]. To this, particular attention must be paid given the analogous long-term effects of TBI, such as recurrent depressive symptoms, dementia, and cognitive impairment, which could mutually influence each other negatively, worsening the overall clinic of the affected subjects [54,55,56].

For these adverse effects, ketamine should be cautiously associated with other mood and perception-altering drugs, including opioids and benzodiazepines. Analogously, because the cytochrome P450 enzyme is involved in ketamine metabolism [50], drugs that inhibit cytochrome P450 metabolism should be avoided for the risk of leading to supra-therapeutic ketamine toxicity. However, overdose death is rare and usually involves other substances [51].

Ketamine in an acute setting could not impact cerebral autoregulation: a delicate mechanism altered in acute brain injury [57].

## 4. Conclusions

As a result of its pharmacokinetic and pharmacodynamic characteristics, including neuromodulation properties, ketamine appears to be a safe drug and could be used alone or in combination with other sedatives in patients with moderate-to-severe TBI requiring mechanical ventilation.

After more than 50 years of research, ketamine use in patients with acute brain trauma still appears to be underused. Various prospective and retrospective trials have been completed, but all of these show a weakness that does not allow for solid recommendations to be formulated. However, no studies have shown any dangers of using ketamine in head trauma. This allows, therefore, to imagine the possibility of including this therapy in well-established clinical practice procedures for the treatment of brain injury. To this end, the next desirable move would be to carry out a double-blind, randomized, controlled multicenter study, identifying the multiple confounding factors by a multidisciplinary team that involves at least an anesthetist, a neurologist-neurophysiopathologist, a toxicologist, and a medico-legal expert.

## Figures and Tables

**Table 1 healthcare-10-00566-t001:** Main results from the literature review.

Intracranial Pressure, Cerebral Perfusion Pressure, Mean Arterial Pressure, Heart Rate	Spreading Depolarization and Burst Suppression	Ketamine Dosage	Ventilation and Arterial CO_2_	Ketamine and Children	Ketamine Toxicity
Aroni et al. [21]Carlson et al. [22];Bourgoin et al. [23];Albanèse et al. [24];Bar-Joseph et al. [25];Hertle et al. [26];Kolenda et al. [27];Zeiler et al. [28];Caricato et al. [29];Bourgoin et al. [30];Schmittner et al. [31];	Stevens et al. [17];Hertle et al. [18];Albanèse et al. [24];Hertle et al. [26];Carlson et al. [32];Akeju et al. [33];Vlisides et al. [34]Opdenakker et al. [35]	Kramer et al. [16];Hertle et al. [18];Bourgoin et al. [26];Hertle et al. [26];Kolenda et al. [28];Caricato et al. [29];Bourgoin et al. [30];Schmittner et al. [31];Carlson et al. [32];Green et al. [33];Akeju et al. [34];Opdenakker et al. [35];Grathwohl et al. [36];Colton et al. [37];Statler et al. [38]	Opdenakker et al. [35];Flower et al. [39];Oddo et al. [40];Hughes [41];Freeman [42];Pfenninger et al. [43]	Bar-Joseph et al. [25];Green at al. [44]Clausen et al. [45];Li et al. [46]	Krystal et al. [47];Jentsch et al. [47];Honey et al. [48];Liu et al. [49];Mössner et al. [50];Orhurhu et al. [51]

**Table 2 healthcare-10-00566-t002:** Main ketamine effects from the systematic review.

Ketamine Effect
**ICP**	-no pathological ICP [23,28,29,30,31];-prevent cough reflex [21];-decreased ICP values in a short time and increased values over a long time without evidence of threat [24];-decrease in median 7.8 mmHg ICP in pediatric population study [25];-no adverse events related to elevated ICP or rising mortality rates [23,24,28,29,30,31].
**CPP**	-CPP increase of 3.9 mmHg without adverse effects [25]
**MAP and HR**	-no notable variation in MAP (mean arterial pressure) and HR (heart rate) [22,23,24,26,29,30];-increase ofMAP [25];-increase in HR of 20 beats/min [27]
**EEG**	-“gamma burst” pattern at the standard ketamine dosage required to induce unconsciousness [33,34];-significantly reduces Spreading Depolarization [SD], with a dose-dependent mechanism [17,18];-ketamine increases the presence of beta frequencies, consequently, limiting the appearance of SD [26];-burst suppression with a dose-dependent relationship [24];-dose-dependent inhibitory effect of ketamine for the incidence of SD [32]
**Ventilation**	-bronchodilatation [13,39];-facilitate airway management in patients with traumatic brain injury [40]

## Data Availability

The data presented in this study are available in references.

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
