# Peer review of "Ketamine in Acute Brain Injury: Current Opinion Following Cerebral Circulation and Electrical Activity"

_healthcare, 2022, doi:10.3390/healthcare10030566_

Round 1

Reviewer 1 Report

Authors should have mentioned about antidepressant effect of ketamine. Only adverse psychical events were taken into account - however it would be useful to write something about depressive symptoms after head injuries.

Maybe some biases should be also mentioned.

Author Response

Thank you to the reviewer for the suggestions. We added notices about antidepressant effect of ketamine and we added some informations on depressive symptoms after head injuries. We also highlighted some other biases of the study.

Reviewer 2 Report

Manuscript needs revision for grammar, spelling and misuse of words.

Need more information regarding mechanism of action of Ketamine in relation to benefit/risk in the categories listed in the manuscript. 

Use of more data from studies cited. 

Author Response

Thank you to the reviewr for his suggestions that will make paper more readeble.

We performed a deep revision of english language and grammar mistakes.

We added more informations about mechanism of action of Ketamine with regard to the benefit/risk in the categories listed in the manuscript and added some more data from the papers we cited in the manuscript. 

Reviewer 3 Report

The review covers an important topic of utilizing ketamine in brain injury treatment. However, there could still be room for improvement.

  1. English editing is needed because there are various unclear sentences. 

"After nine years until nowadays, ketamine got in operative room as a general anesthetic in humans "

"After binding with NMDA receptor, the main clinical effect is dissociative without 50 vascular bed dilatation, hypotension, and heart chronotropic effect."

"The caution about the use of a ketamine in patients with TBI is due to the possibility 61 that ketamine may increase intracranial pressure through sympathetic stimulation, worsening the outcome of the clinical condition."

  1. Many obscure terms are used. 
  • It is also not clear what the authors mean by "electric activity complex".
  • It is not clear what the authors mean by "In three studies, no differences of the ICP between the patients who received ketamine and the control group and none shown pathological ICP [24, 28-31]."

There are plenty more unclear sentences. 

The review could be considered to be a little underdeveloped, and require further development and enhancement. For example, complete paragraphs could be added. A clear introduction is still needed, also clear findings of the metanalysis and clear conclusion. The review needs figures that summarise the idea and more tables to support the findings.

Author Response

Thank you to the reviewer for his suggestions.

We performed a deep revision of grammar and tried to be clearer in some sentences.

According to the reviewer's suggestions we tried to add some more development and enhancement to do the review more complete adding complete paragraphs and adding an introduction.

As suggested we added some figures and tables to support the findings.